Cultural sensitivity; Digital mental health; Forced migration; Mental illness; Mobile health

**Author for correspondence:**
Laura Nohr,
Email: laura.nohr@fu-berlin.de

# Smartphone-delivered mental health care interventions for refugees: A systematic review of the literature

Rayan El-Haj-Mohamad[1,2], Laura Nohr[1] 🔘, Helen Niemeyer[1], Maria Böttche[1,2] and Christine Knaevelsrud[1]

[1]Division of Clinical Psychological Intervention, Department of Education and Psychology, Freie Universität Berlin, Berlin, Germany and [2]Center Überleben, Berlin, Germany

## Abstract

According to the United Nations, an estimated 26.6 million people worldwide were refugees in 2021. Experiences before, during, and after flight increase psychological distress and contribute to a high prevalence of mental disorders. The resulting high need for mental health care is generally not reflected in the actual mental health care provision for refugees. A possible strategy to close this gap might be to offer smartphone-delivered mental health care. This systematic review summarizes the current state of research on smartphone-delivered interventions for refugees, answering the following research questions: (1) Which smartphone-delivered interventions are available for refugees? (2) What do we know about their clinical (efficacy) and (3) nonclinical outcomes (e.g., feasibility, appropriateness, acceptance, and barriers)? (4) What are their dropout rates and dropout reasons? (5) To what extent do smartphone-delivered interventions consider data security? Relevant databases were systematically searched for published studies, gray literature, and unpublished information. In total, 456 data points were screened. Twelve interventions were included (nine interventions from 11 peer-reviewed articles and three interventions without published study reports), comprising nine interventions for adult refugees and three for adolescent and young refugees. Study participants were mostly satisfied with the interventions, indicating adequate acceptability. Only one randomized controlled trial (RCT; from two RCTs and two pilot RCTs) found a significant reduction in the primary clinical outcome compared to the control group. Dropout rates ranged from 2.9 to 80%. In the discussion, the heterogeneous findings are integrated into the current state of literature.

## Impact statement

Refugees are a large population with special mental health care needs which are nowadays not adequately addressed by most of the host countries. Experiences before, during, and after flight increase psychological distress and contribute to a high prevalence of mental disorders. The resulting high need for mental health care is generally not reflected in the actual mental health care provision for refugees. Potential reasons for low utilization include language difficulties, limited treatment offer, and lack of knowledge about mental health care systems. A possible strategy to close this gap might be to offer smartphone-delivered mental health care. Since most refugees own a smartphone, the smartphone represents a great health care opportunity. The current systematic review gives an overview of the existing stand-alone smartphone-delivered interventions for mental health problems in refugee populations. We identified nine interventions for adults and three for adolescents and young refugees. The review enables the audience to identify treatments in different languages, targeting different mental problems, and offering varying amounts of support. This helps persons affected, persons working with refugee populations, and stakeholders to identify the most fitting interventions for specific persons or populations. In the course of the summarized trials, about 400 refugees were provided with smartphone-delivered mental health care. The results show that the different interventions were able to improve single aspects of mental health and well-being. Still, we identified room for improvement in the efficacy and effectiveness of smartphone-delivered interventions, the involvement of post-migration stressors in the treatment, and data safety. This knowledge helps scientists and stakeholders to decide which steps should be taken next to fully exploit the potential of smartphone-delivered mental health interventions for refugees. For instance, we need more knowledge about effective treatment elements, facilitating characteristics to improve their use, and barriers that hamper the wide use in refugee populations.





## Introduction

According to the United Nations, there were an estimated 84 million forcibly displaced people worldwide in 2021 (UNHCR, 2022a). Political developments and armed conflicts, as well as climate change, have recently led even more people to leave their home countries to seek asylum elsewhere (e.g., UNHCR, 2022b,c). Experiences before, during, and after flight increase psychological distress and the risk of various mental disorders, with a recent umbrella review identifying depression, anxiety, and post-traumatic stress disorder (PTSD) as the most common mental disorders among refugees, accounting for up to 40% of all mental disorders in this population (Turrini et al., 2019). Additional post-migration stressors frequently experienced in the host countries (e.g., discrimination, poor living conditions, new cultural context, and language barriers) pose further challenges for refugees and are also associated with worse mental health (Tinghög et al., 2017; Malm et al., 2020). In general, refugees show a low level of well-being (Leiler et al., 2019; Beza et al., 2022). Despite the correspondingly high need for diverse psychosocial interventions and treatment, the actual mental health care and its use in the host countries is low (Satinsky et al., 2019). Potential reasons for low utilization include structural barriers (e.g., language and cultural barriers, and lack of health care options; Kiselev et al., 2020a) and personal barriers such as mental health stigma, avoidance of symptoms, and limited mental health literacy (Shannon et al., 2015; Kiselev et al., 2020b). Moreover, at first sight, adequate mental health care for refugees might appear burdensome for the host countries and its practitioners (e.g., due to additional costs for interpreters; Gadon et al., 2007). Thereby, benefits and lower costs in the long term are frequently not considered (Brandl et al., 2020). Therefore, there is an urgent demand for innovative treatment options that address both the diverse needs of refugees as well as cost-effectiveness and scalability for the host countries.

### *Smartphone-delivered mental health interventions*

Smartphone-delivered interventions may be a promising approach to close this treatment gap for refugees. Since most people, and especially refugees, own a smartphone rather than other digital devices, the smartphone represents a great health care opportunity (Casswell, 2019). Smartphone-delivered interventions like applications ('apps') or internet-based interventions that can be implemented on smartphones have the potential to directly address some of the strongest barriers to help-seeking in refugee populations. Such interventions are flexible in terms of time and place, can be offered in different languages, thus reaching a high number of people, and their anonymous usage might reduce the fear of mental health stigma (Hilty et al., 2018; Burchert et al., 2019; Schmidt-Hantke et al., 2021).

While these advantages of smartphone-delivered interventions seem compelling, there is a need to investigate their effectiveness, (cost-)efficiency, usability, and acceptability in refugee populations. Currently available smartphone-delivered interventions differ widely in their cultural and contextual adaptations (Spanhel et al., 2021) and in their treatment approach and range of indications (e.g., targeting one specific problem vs. a transdiagnostic approach). Thus, some interventions target psychopathology while others take a more salutogenic approach, with the aim of improving quality of life. Beyond this, interventions can be unguided, offering treatment without personal contact or individualized feedback, or guided, offering varying amounts of personal support (Andersson, 2018;

Bennett et al., 2019). Smartphone-delivered interventions also differ in their design, content presentation (e.g., text- or video-based), usability, and other user experience aspects (cf. Chandrashekar, 2018). Lastly, data security has been identified as an important aspect of refugees' utilization of internet-based mental health interventions (Burchert et al., 2019) and should therefore be particularly emphasized (Liem et al., 2021). All of these characteristics of smartphone-delivered interventions might contribute to differences in their effectiveness and efficacy for specific target populations.

The present review aimed to provide an overview of existing smartphone-delivered mental health interventions, their specific characteristics, and evidence on clinical and nonclinical outcomes explicitly for refugee populations. The latest literature already encompasses some reviews focusing on smartphone-delivered interventions for refugees, but these also included other populations, topics, or internet-based interventions in general. For instance, Wirz et al. (2021) described digital mental health interventions for Arabic- and Persian-speaking persons remaining in their home countries and refugees elsewhere. The authors identified nine app- and web-based mental health interventions for anxiety, depression, and PTSD; two of these were evaluated and one achieved a significant reduction in the primary outcome. Liem et al. (2021) summarized 16 digital mental health interventions for immigrants and refugees. These interventions covered both forcibly and voluntarily migrated populations worldwide and were delivered via all digital devices (e.g., computers) rather than primarily via smartphones. The participants reported general satisfaction and positive attitudes toward digital mental health care interventions, but ethical standards were poorly implemented and reported, and the authors identified mental health stigma and lack of technology literacy as the main challenges. Spanhel et al. (2021) conducted a systematic review on cultural adaptation of internet-based interventions for marginalized groups worldwide, and identified 17 components regarding content, methods, and procedural components eligible for cultural adaptation in the context of internet- and mobile-based mental health interventions.

The aforementioned reviews indicate that refugees can be reached through internet-based and smartphone-delivered interventions. However, to date, no review has included an overview of all existing interventions for refugees that exclusively utilize smartphones.

### *Aims of the systematic review*

The aim of the current systematic review was to provide an overview of existing smartphone-delivered mental health interventions that explicitly address the needs of refugee populations (e.g., dealing with experiences in the home country and during flight, post-migration stressors in the host country). Based on the PICO criteria (*P*opulation – *I*ntervention – *C*omparison – *O*utcome; McKenzie et al., 2019), we identified interventions and study reports targeting forcibly displaced persons of all ages not living in their home countries as the study population. We included published and unpublished information (e.g., peer-reviewed studies, gray literature, and informal communication) on smartphone-delivered mental health interventions. To summarize as much information as possible, we included interventions and research reports on any stage of an intervention's development (e.g., study protocols, feasibility studies, pilot studies, and usability studies) and evidence testing. Interventions to improve any aspect of mental health or

quality of life were included. Specifically, the following research questions were addressed:

1. Which smartphone-delivered mental health interventions are available for refugees?
2. What are the clinical outcomes and efficacy of these smartphone-delivered mental health interventions?
3. What are the nonclinical outcomes of these smartphone-delivered mental health interventions (e.g., feasibility, appropriateness, acceptance, and barriers)?
4. What are the dropout rates and reasons for dropout of the different interventions?
5. To what extent do the smartphone-delivered interventions consider data security?

## Methods

The current systematic review was conducted and reported as recommended by the Preferred Reporting Items for Systematic Reviews and Meta-Analyses (PRISMA) guidelines (Liberati et al., 2009; Page et al., 2021; see Online Supplement 1 of the Supplementary Material for PRISMA checklist). The systematic review was not registered and no register protocol exists. During the process, no modifications were made to the initially agreed search procedure or methods as described below.

### Eligibility criteria

Data points were included if they (a) reported smartphone-delivered interventions aiming to improve mental health or quality of life (b) in refugee populations not living in their home countries. Regarding the study design, (c) primary studies such as randomized controlled trials (RCTs), quantitative, qualitative, or mixed methods, feasibility or pilot studies, and peer-reviewed study protocols as well as unpublished information (d) available between 01/2000 and 04/2022 (e) in the English or German language were considered. We explicitly chose not to restrict the countries of origin and resettlement countries or study participants' age, mental disorders, and symptom severity. Data points were excluded if they (a) did not include smartphone-delivered interventions, for example, tele- or videoconferencing interventions, online assessments and diagnostics, virtual reality (VR), ecological momentary assessments (EMAs), and ecological momentary interventions (EMIs), or were not aimed at improving mental health or quality of life, for example, strengthening social support. Interventions targeting (b) populations like voluntarily migrated persons, second-generation immigrants, internally displaced or indigenous people were excluded. Furthermore, (c) reviews, meta-analyses, commentaries, and (d) data points published or available before 01/2000 or after 04/2022, and only available in (e) languages other than English or German were also excluded.

### Search strategy

To identify eligible articles, two researchers (R.E. and L.N.) independently searched PubMed and the results of the search engine EBSCOhost which was used to simultaneously search the databases CINAHL and MEDLINE with Full Text, APA PsycArticles, and APA PsycInfo. The applied search terms (see Online Supplement 2 of the Supplementary Material) were a combination of relevant keywords related to smartphone-delivered mental health interventions, refugee populations, and various mental health outcomes. At the same time,

the somatic conditions stroke and cancer were explicitly excluded as keywords due to their high coincidence with the search term 'survivor' and a consequently large number of findings not fitting the scope of the review. The search was limited by applying filters on publication date and type of study report. The literature search was realized on April 30, 2022. To further reduce potential bias, additional search strategies were applied, and gray literature not previously peer-reviewed (see Conn et al., 2003) and unpublished information were identified. Therefore, we contacted leading experts in the field of digital and smartphone-delivered mental health, psychological treatment for refugee populations, and transcultural clinical psychology. Next, citation searching was applied and the reference lists of all included studies, previous systematic reviews, and systematic reviews on related topics were systematically searched. Finally, additional databases for preprints (PsyArXiV and OSF), clinical registrations (US and European clinical register), and conference volumes of the WCCBT 2019 (Heidenreich and Tata, 2019; Heidenreich et al., 2019) and the Swedish Congress on Internet Interventions 2022 (Andersson et al., 2022) were systematically searched. Identified studies and data sources were only included if published or available before April 30, 2022.

### Selection process

All references from PubMed and EBSCOhost were imported into the online open-source software *Rayyan* (Ouzzani et al., 2016) for initial screening of titles and abstracts. *Rayyan* automatically identified potential duplicates, which we checked and removed manually where necessary. Subsequently, R.E and L.N. screened titles and abstracts independently. Screening followed a hierarchical approach, applying the inclusion and exclusion criteria as presented in the Online Supplement 3 of the Supplementary Material. Next, initially included articles, gray literature, and unpublished data sources were screened based on their full texts by R.E and L.N. independently, which resulted in the ultimately included data points. Disagreements regarding inclusion were discussed with a third researcher (M.B.) and resolved by consensus. To control for interrater reliability, Cohen's $\kappa$ was calculated (Cohen, 1960). For a detailed description of the selection process, see the PRISMA flow diagram in Figure 1 and Online Supplement 3 of the Supplementary Material.

### Quality assessment

For quality assessment, studies were grouped by study type. Only data points offering any type of study report were rated regarding general quality aspects. Unpublished and informal information was not assessed. The quality assessment sought to structure the systematic evaluation of included studies and support the systematic identification of strengths and weaknesses of published studies in this field. To adequately address the diversity of included studies, each study type was evaluated by a well-established and standardized quality assessment tool identified by the Equator network (2021). For RCTs, the CONSORT checklist was applied (Schulz et al., 2011); for pilot RCTs and feasibility studies, the CONSORT 2010 extension was used (Eldridge et al., 2016); for study protocols, the Standard Protocol Items: Recommendations for International Trials (SPIRIT) was applied (Chan et al., 2013); and for qualitative studies, the Standards for Reporting Qualitative Research (SRQR) were used (O'Brien et al., 2014). All quality assessments were undertaken individually by R.E. and L.N. Observed agreement ($P_0$) was calculated by dividing the number of agreements by the total number for each data extraction item (Cohen, 1960).

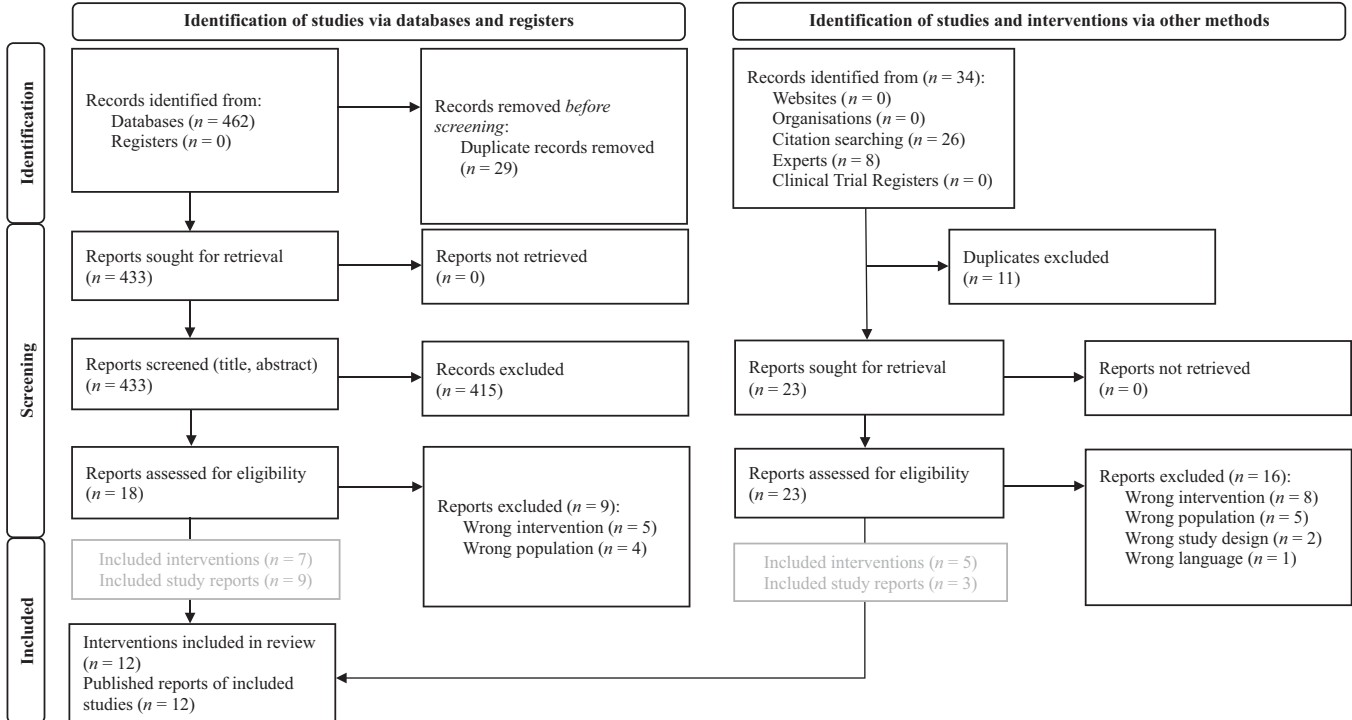

**Figure 1.** PRISMA 2020 flow diagram for the systematic review about the searches of databases, registers, and other sources (Page et al., 2021).

### Data extraction

To answer the *a priori* defined research questions, relevant information was extracted for each intervention and study report (see Table 1). First, characteristics of each intervention were extracted: name and principal aim, language(s), length in modules and planned duration in weeks or months, type (guided vs. unguided), and adaptation (cultural and contextual). Next, respective study reports associated with each intervention were reported with the following information: first author and year of publication, aim of the study, research design used, primary clinical outcome (if available), reported nonclinical outcome, log data, and dropout rates. Data extraction was conducted individually by R.E. and L.N. following a standardized template and observed interrater agreement ($P_0$) was assessed (Cohen, 1960). Disagreements were resolved by consensus.

Furthermore, a special emphasis was placed on data security in the context of smartphone-delivered interventions. Given the growing importance of data security (Gaebel et al., 2021), especially in refugee populations (Burchert et al., 2019), the review aimed, from an exploratory perspective, to extract information on data security characteristics of smartphone-delivered mental health interventions and associated research studies, for example, where and for how long the data are stored, who has access to data, among others.

### Results

The literature search yielded a total of 462 records. After removing duplicates, 433 titles and abstracts were screened. Eighteen full texts were considered for full-text analyses. Nine of these full texts describing eight different interventions met the inclusion criteria and were included in the review. A further three full texts describing two novel interventions were identified through citation search and additionally included. No gray literature was eligible for inclusion. Three interventions without any published or unpublished reports were

found to be eligible to answer the first research question. These interventions were identified by experts and were therefore included. No further information on their use, clinical and nonclinical outcomes, or drop-out rates can be provided. In total, nine smartphone-delivered mental health interventions for adult refugees and three for adolescent and young refugees were identified (see Figure 1). Inter-rater reliability for the literature search was $\kappa = .68$, indicating a substantial interrater agreement (Landis and Koch, 1977).

### Characteristics of included interventions

The identified interventions offer a wide variety of different treatment approaches (see Tables 1 and 2). A total of 5/12 interventions explicitly address depressive symptoms (*Almamar*, 'iCBT', 'iCBT youth', iFight Depression, and Step-by-Step), 3/12 of which additionally target anxiety symptoms (*Almamar*, 'iCBT', and 'iCBT youth'). A total of 2/12 interventions explicitly target the treatment of PTSD (*Almamar* and Sanadak) and 2/12 seek to improve only specific symptoms such as sleep problems and concentration difficulties due to intrusive memories (eSano Sleep-e and 'Tetris'). A total of 1/12 interventions try to improve mental health outcomes indirectly by addressing mental health care stigma and help-seeking attitudes (Tell Your Story). A total of 2/12 interventions focus on addictive behavior and substance abuse (*Almamar* and BePrepared). The treatment approaches also differ regarding psychotherapeutic guidance: 7/12 are unguided (ALMHAR, Balsam, BePrepared, eSano Sleep-e, Sanadak, Tell your Story, and 'Tetris'), 3/12 are guided ('iCBT', 'iCBT youth', and iFight Depression), 1/12 are minimally guided (Step-by-Step), and 1/12 offer both a guided and an unguided version (*Almamar*). A total of 1/12 interventions are based on gamification linked to an instruction to remember traumatic events ('Tetris'). In another intervention (1/12), a short self-test on post-traumatic symptom severity is implemented to allow for automated tailored feedback regarding progress at any

**Table 1.** Smartphone-delivered mental health interventions for refugees and characteristics of associated publications

| Name and aim of intervention | Language | Type/length | Adaptation | Study reports (author, year) | Aim of the study | Research design/ sample size (% f/m/d) | Primary clinical outcome/efficacy | Nonclinical outcome | App use/drop out |
|---|---|---|---|---|---|---|---|---|---|
| *Balsam* Psycho-education and coping skills to deal with distress based on a visualized story-telling approach | Arabic, English, Farsi, German | Unguided 15 modules (12 weeks) | *Content*: Stigma, symptom manifestation, cultural belonging, acculturation and explanatory models of mental illness *Context*: Developed for online use for refugee populations *Culture*: Based on Arabic literature, culture, and explanatory models of mental illness; including acculturation | Böge et al. (2020) | Efficacy testing of the Stepped Care and Collaborative Model (Härter et al., 2018) | Study protocol Multicentric RCT Stepped care vs. TAU (active control) $N_{planned}$ = 276 | Depression | Cost effectiveness, acceptability, credibility, and expectancy of treatment | NA NA |
| *BePrepared* Indicated prevention to reduce problematic use of alcohol and cannabis in young adults (based on CBT principles) | Arabic, English, Farsi, German, Pashto | Unguided 4 main modules + 4 optional modules (4 weeks) | *Content*: Self-monitoring, motivation to change, skills focusing on behavior, thoughts, and emotions, psychoeducation on safe handling of alcohol and cannabis, activation of personal resources, and information on the addiction help system in Germany *Context*: Developed as a smartphone-delivered intervention *Culture*: Culturally sensitive adaptation following the guidelines of Bernal and Sáez-Santiago (2006) | Fischer et al. (2021) | Feasibility, usability, and acceptability testing | Study protocol Single-armed feasibility trial $N_{planned}$ = 150 | Change in substance use post-intervention | Intervention's feasibility and acceptance in the target population | NA |
| *eSano Sleep-e* Transdiagnostic intervention to reduce sleeping problems based on highly structured CBT-interventions for insomnia | English, German | Unguided 4 modules (30–45 min each) (4 weeks) | *Content*: Psychoeducation on sleep problems, sleep hygiene, sleep medication, and problems related to sleep; relaxation exercise, sleep diary; exercises to deal with rumination and sleep hygiene, additional resources *Context*: Adapted from a digital intervention for German teachers for the smart-phone use *Culture*: Based on | Spanhel et al. (2019) | Identification of interventions' elements in need of cultural adaptation | Qualitative study on cultural adaptation $N$ = 6 refugees (16.7/83.3/NA) $N$ = 6 mental health care providers (66.7/33.3/NA) | NA | Consideration of refugees' characteristic: Problems and stressors, everyday habits, socialization, values Disease and treatment concepts Adaptation of the intervention: Pictures, role models, language, psychoeducational elements, structure of | NA NA |

(Continued)

| Name and aim of intervention | Language | Type/length | Adaptation | Study reports (author, year) | Aim of the study | Research design/ sample size (% f/m/d) | Primary clinical outcome/efficacy | Nonclinical outcome | App use/drop out |
|---|---|---|---|---|---|---|---|---|---|
| | | | heuristic framework adaptations of content, language, and methods | | | | | modules, format of presentation | |
| | | | | Spanhel et al. (2022) | Preliminary effectiveness, feasibility, and acceptability testing of the intervention to reduce sleeping problems | Pilot RCT IG vs. WLC $N = 66$ (27.3/72.3/NA) | No between-mean group differences and group × time interaction of insomnia severity Mean-difference = −2.1 (95% CI: −4.8–0.5), $p = .112$ Hedge's $g = .40$ (95% CI: −.09–.88) | High satisfaction with the intervention and its perceived cultural appropriateness Negative effects: Worsened symptoms; deficiencies of the intervention; dependency; stigma, hopelessness | $M = 2.9$ ($SD = 1.7$) modules completed (=72.0% of the intervention) 66.7% completed all modules *Dropout*: IG: 18.2% CG: 15.2% |
| *'iCBT'* Intervention to reduce symptoms of anxiety and depression for adults (≥18 years) based on CBT-interventions | Arabic | Guided (individualized) 9 modules (8 weeks) | *Content*: CBT-interventions targeting anxiety, depression, insomnia, stress, emotion regulation, worry and intrusive memories/flash backs *Context*: Adapted from a digital intervention for Arabic speaking refugees *Culture*: Adaptations focused on making the material, the language, and case examples easily accessible for persons from varying cultural backgrounds | Lindegaard et al. (2021b) | Efficacy testing in reducing symptoms of anxiety and depression | Pilot RCT IG vs. WLC $N = 59$ (42.0/58.0/NA) | Significant group × time effect on depressive symptoms Mean difference = −0.42 (95% CI −0.82–−0.02, $z = −2.06$, $p = .039$) Cohen's $d = 0.85$ (95% CI 0.29–1.41) | No negative effects of the intervention reported in the IG | $M = 2.23$ modules completed *Dropout*: IG: 40.0% CG: 37.9% |
| | | | | Lindegaard et al. (2021a) | User experience testing | Qualitative feasibility study $N = 10$ (60.0/40.0/NA) | NA | 1. The importance of being seen 2. New ways of knowing and doing 3. Treatment format not for everyone 4. Changing attitudes toward mental health and help-seeking 5. The healthcare system as a complex puzzle | NA NA |
| *'iCBT youth'* Intervention to reduce symptoms of anxiety and depression for | Dari, Farsi | Guided (individualized) 9 modules (8 weeks) | *Content*: See above *Context*: Adapted from original Arabic iCBT intervention for adults *Culture*: Adaptations focused on linguistic | Lindegaard et al. (2022) | Feasibility and acceptability testing | Qualitative feasibility study Intention to treat sample: $N = 15$ | Planned quantitative analyses not realized due to high dropout rates | Very low feasibility, acceptability, and adherence *Barriers*: Cultural differences, internal circumstances | $M = 0.9$ modules completed *Dropout*: 80.0% IG |

**Table 1.** (*Continued*)

| Name and aim of intervention | Language | Type/length | Adaptation | Study reports (author, year) | Aim of the study | Research design/ sample size (% f/m/d) | Primary clinical outcome/efficacy | Nonclinical outcome | App use/drop out |
|---|---|---|---|---|---|---|---|---|---|
| adolescents and young adults (15–26 years) | | | translation and simplifying the language; adding modules about prolonged grief and separation anxiety | | | (93.0/7.0/NA) Qualitative interviews: n = 4 (0.0/100.0/NA) treatment participants n = 3 (33.3/66.7/NA) none treatment participants | | (interfering symptoms, low trust), external circumstances, treatment (lack of human contact, technical difficulties, content not relevant) *Facilitators*: Easy to understand, helpful content, intuitive platform, online format | |
| *Sanadak*[1] Interactive intervention developed based on evidence-driven CBT for PTSD | Arabic | Unguided NA (4 weeks) | *Content*: Psychoeducation on PTSD, related mental health issues, and self-help techniques; skills training for symptom management; self-test; interactive materials *Context*: Developed as a smartphone app; including video, audio sequences, interactive games, and exercises *Culture*: Culturally sensitive adaptation in collaboration with peers with Arabic background, e.g., disease and disease management | Röhr et al. (2021) | Testing of effectiveness in reducing PTSD symptoms and evaluating cost effectiveness | RCT IG vs. active CG N = 133 (61.7/38.3/NA) | No significant differences in PTSD symptoms between the IG and CG after 4 weeks, mean-difference = −0.90 (95% CI −0.24–0.47, p = .52) and after 4 months, mean-difference = −0.39 (95% CI −3.2–2.46; p = .79) | No cost effectiveness | M = 42.5 min app use *Dropout*: IG: 9.2% CG: 2.9% |
| *Step-by-Step* Intervention to reduce symptoms of depression based on WHOs evidence-based Problem Management Plus (PM+) program | Arabic | Minimally guided 5 modules (30min each) | *Content*: Based on CBT techniques (behavioral activation, psychoeducation, stress management, increasing social support and relapse prevention) using story telling through illustrated educative narratives and interactive exercises presented by a fictional main character and a fictional health professional *Context*: Smartphone app adapted from an online intervention; for example, | Burchert et al. (2019) | Adaptation of the intervention to the needs and expectations of Syrian refugees using early prototyping and usability testing | Qualitative study N = 128 n = 60 (n = 33 analyzed) free list interviews n = 36 key informant interviews n = 32 participants in focus groups (50.0/50.0/NA) | NA | Overall positive reaction on usability Easy to use and to understand Flexibility and customizability Length and pace of sessions were criticized *Barriers*: Technology literacy and requirements, acceptance of having psychological problems, lack of trust, acceptability, credibility, too | NA NA |

**Table 1.** (*Continued*)

| Name and aim of intervention | Language | Type/length | Adaptation | Study reports (author, year) | Aim of the study | Research design/ sample size (% f/m/d) | Primary clinical outcome/efficacy | Nonclinical outcome | App use/drop out |
|---|---|---|---|---|---|---|---|---|---|
| | | | videos, illustrations *Culture*: Culturally sensitive content, illustrations, and avatars | | | | | distressed *Facilitators*: Promotion of the app | |
| *Tell Your Story* Evidence-based intervention to reduce mental health self-stigma and increase help-seeking in men with symptoms of PTSD | Arabic, Farsi, Tamil | Unguided 11 modules (4 weeks) | *Content*: Social contact, psycho-education, and cognitive reappraisal to specifically target self-stigma related to PTSD and help-seeking *Context*: Developed as online intervention for refugee men *Culture*: Culturally sensitive adaptation in collaboration with peers with Arabic, Farsi, and Tamil background, e.g., gender sensitivity, material, and strategies | Nickerson et al. (2019) | Effectiveness of the intervention regarding mental health self-stigma, help-seeking, and mental health outcomes | RCT IG vs. WLC $N = 103$ (0.0/100.0/0.0) | No significant effects of time, condition or time × condition interaction for PTSD symptoms | No significant effects for self-stigma for PTSD No significant association between the number of modules completed and self-stigma for PTSD, self-stigma for seeking help or help-seeking IG: Less increase in self-stigma for help-seeking; decrease in help-seeking intentions; more help-seeking from new sources CG: Increase in self-stigma for help-seeking Usability: Program easy to use and understand, attractive, useful, interesting, engaging, and content was relevant and useful | $M = 4.76$ ($SD = 3.86$) modules completed *Dropout*: IG: 18.5% CG: 18.4% |
| *'Tetris'* Game to reduce frequency of intrusive memories of trauma for adolescents and young adults (16–25 years) | Arabic instruction | Unguided 1 session (15 min) | *Content*: Instruction to think about a traumatic memory and to play the game Tetris *Context*: — *Culture*: — | Holmes et al. (2017) | Acceptability, feasibility, and preliminary effectiveness of a brief cognitive science-driven intervention without interpreters | Qualitative feasibility study $N = 22$ (22.7/77.3/NA) | NA | Overall positive feedback on acceptability and feasibility (e.g., easy to use, enjoyable) Reasons for dropout: Instructions not understood, too sad to complete task because of bad news | $M = 18.0$ ($SD = 2.4$) minutes game playing *Dropout*: 22.7% IG |

*Note*: CBT, cognitive behavioral therapy; CG, control group; IG, intervention group; *M*, mean; min, minutes; NA, not available; PTSD, post-traumatic stress disorder; RCT, randomized controlled trial; SD, standard deviation; TAU, treatment as usual; WLC, wait list control group.

[1]The systematic literature search identified a peer-reviewed study protocol of the planned RCT (Golchert et al., 2019). Since the trial has been completed at the time of the systematic review, we decided to not extract information from the study protocol to avoid redundancy.

**Table 2.** Interventions without published information.

| Name and aim of intervention | Language | Type/length | Adaptation | Current state of research |
|---|---|---|---|---|
| *Almamar*<br>Individually tailored intervention addressing post-traumatic, depressive, anxiety symptoms, and addictive behavior | Arabic, Farsi | Guided/unguided 32 modules (6–16 weeks) | *Context*: Smartphone app adapted from a face-to-face intervention<br>*Culture*: Culturally adapted | The app is undergoing feasibility testing<br>A RCT and effectiveness testing is planned for Arabic- and Farsi-speaking refugees |
| ALMHAR (AppLication for Mental Health Aid for Refugees)<br>Psychoeducational app to reduce distress | Arabic, English, Farsi | Unguided 12 modules | *Context*: Developed as an internet-based intervention<br>*Culture*: Culturally adapted from an international version | No data on clinical and nonclinical outcomes available |
| iFight Depression<br>Intervention to reduce depressive symptoms | Arabic, German, among others | Guided<br>— | *Context*: Developed as an internet-based intervention<br>*Culture*: Culturally adapted from an international version | The intervention is undergoing log data analysis of the routine care use |

*Note:* RCT, randomized controlled trial.

time (Sanadak). To maximize usability, Sanadak also provides interactive materials such as animated video and audio as well as games and exercises. No other interventions used gamification. Regarding language, 8/12 interventions are offered in several languages (*Almamar*, ALMHAR, Balsam, BePrepared, eSano Sleep-e, 'iCBT youth', iFight Depression, and Tell Your Story), 10/12 in Arabic (*Almamar*, ALMHAR, Balsam, BePrepared, 'iCBT', iFight Depression, Sanadak, Step-by-Step, Tell Your Story, and 'Tetris'), 6/12 in Farsi (*Almamar*, ALMHAR, Balsam, BePrepared, 'iCBT youth', and Tell Your Story), 1/12 in Dari ('iCBT youth'), 1/12 in Pashto (BePrepared), and 1/12 in Tamil (Tell Your Story). Additionally, 1/12 interventions are offered in easily understandable English and German (eSano Sleep-e). Beyond language, all interventions but one ('Tetris') were culturally adapted to fit refugee populations or were explicitly developed for this purpose (11/12). Regarding context, 2/12 interventions were originally developed as smartphone-based interventions (BePrepared and Sanadak), 7/12 are internet-based or adapted from internet-based to smartphone-delivered interventions (ALMHAR, Balsam, eSano Sleep-e, 'iCBT youth', iFight Depression, Step-by-Step, and Tell Your Story), and 2/12 were based on face-to-face interventions (*Almamar*) or offline self-help material ('iCBT'). Although all included interventions explicitly aim at improving mental health for refugee populations, none reported content or interventions on post-migration stressors.

Furthermore, the identified interventions were at different stages of their evaluation process. We included two RCTs (Nickerson et al., 2019; Röhr et al., 2021) and two pilot RCTs (Lindegaard et al., 2021b; Spanhel et al., 2022), three feasibility studies (Holmes et al., 2017; Lindegaard et al., 2021a, 2022), two qualitative studies on the development process (Burchert et al., 2019), and the cultural adaptation of interventions (Spanhel et al., 2019). Moreover, two peer-reviewed study protocols (Böge et al., 2020; Fischer et al., 2021) and three interventions without any published study reports were identified. Of the latter, two were identified by experts (*Almamar* and iFight Depression) and one by snowballing (ALMHAR).

### Clinical and nonclinical outcomes of interventions

According to the respective stage of evaluation, different clinical and nonclinical outcomes were reported: Four of the identified study reports collected data on clinical outcomes and tested efficacy in the context of RCTs (Nickerson et al., 2019; Röhr et al., 2021) or pilot RCTs (Lindegaard et al., 2021b; Spanhel et al., 2022). One of the included study protocols described a planned RCT on the efficacy of a stepped and collaborative care model including the smartphone-delivered intervention as one of several low-threshold stand-alone interventions (Böge et al., 2020). The other study protocol described a planned one-armed feasibility and acceptability trial of an app targeting problematic use of alcohol and cannabis (Fischer et al., 2021). Although one feasibility study also collected data on clinical outcomes, no pre–post comparisons or other inferential statistics were reported (Holmes et al., 2017).

One study reported a significant difference between the intervention and control group in the pre–post comparison (Lindegaard et al., 2021b). Three studies did not report significant results regarding their primary outcomes (Nickerson et al., 2019; Röhr et al., 2021; Spanhel et al., 2022). Additionally, most of the studies reported several secondary clinical outcomes, with only few significant results. At the same time, the control conditions in the studies differed greatly: Three studies applied waitlist control conditions (Nickerson et al., 2019; Lindegaard et al., 2021b; Spanhel et al., 2022), one used an active control condition (Röhr et al., 2021), and the study protocol reported a planned treatment-as-usual control condition (Böge et al., 2020).

Dropout rates also varied widely between studies. While most reported low dropout rates (2.9–18.5%; Nickerson et al., 2019; Röhr et al., 2021; Spanhel et al., 2022), some reported high rates (37.9–80%; Lindegaard et al., 2021b, 2022). Lastly, few participants worked on all modules offered throughout the interventions or used the interventions as recommended. On average, participants completed 37.51% of the respective intervention (10–72%; Lindegaard et al., 2022; Spanhel et al., 2022). Overall, most study reports lack clear information about the recommended dose of intervention.

The included studies reported a broad variety of nonclinical outcomes. In particular, the qualitative studies identified relevant themes regarding feasibility, usability, treatment barriers, and content. Six study reports included statements about the acceptance of the respective intervention (Holmes et al., 2017; Burchert et al., 2019; Nickerson et al., 2019; Lindegaard et al., 2021b, 2022; Spanhel et al., 2022). With the exception of one study (Lindegaard et al.,

2022), the findings indicated a general acceptance of the interventions by refugees. Nevertheless, some aspects were identified as conflicting. For instance, some participants mentioned anonymous participation as a major benefit of smartphone-delivered interventions, while at the same time, participants wished for more personal contact like regular telephone calls or face-to-face meetings with their therapist (Burchert et al., 2019). Identified treatment barriers included lack of technological literacy and respective difficulties, cultural differences, lack of trust in data security, and written-based treatment. One study summarized that participants dropped out because they did not understand the instructions or were too sad to complete the task (Holmes et al., 2017).

### Data security

Although lack of data security and a consequent lack of trust in smartphone-delivered interventions was identified as an important barrier, only some of the identified reports addressed this topic. Therefore, we excluded this information from our data extraction table. Five study reports provided information on the topic of data security: one RCT (Röhr et al., 2021), one pilot RCT (Lindegaard et al., 2021b), one qualitative usability study (Burchert et al., 2019), and the study protocols (Böge et al., 2020; Fischer et al., 2021). The qualitative study identified data security as important for the target group. The RCT fulfilled high standards regarding European data security policies, while the study protocols described an overall elaborated data security concept involving specialist lawyers. Finally, the pilot RCT referred to an external webpage called *Iterapi*, which guarantees data security (Vlaescu et al., 2016). All other reports did not mention data security measures at all. Interrater agreement for data extraction ranged from 75 to 100% (see Online Supplement 4 of the Supplementary Material). Lower agreement might have resulted from a less detailed description of some aspects in single study reports.

### Quality appraisal

The quality of studies was rated using the instruments described in the 'Methods' section. These enable the rating of the availability of quality criteria regarding title and abstract, introduction, methods, results, discussion, and other information like funding or competing interests. Due to small numbers of each study type, RCT and pilot RCT quality assessments were summarized based on the similarity of ratings. Most of the studies fulfilled the proposed quality criteria of the CONSORT checklists (Schulz et al., 2011; Eldridge et al., 2016). However, single aspects were not addressed with sufficient detail to allow future researchers to replicate the study design, for example, recruitment strategies were reported only superficially. Moreover, sample sizes were overall small, ranging from $N = 59$–133, which does not allow for a generalization of findings and conclusions. Additionally, the samples were only partly representative. In most samples, male refugees aged 28–41 years were overrepresented, so that these results cannot be generalized to female or gender-diverse refugees or other cultural refugee groups.

Quality assessment of qualitative studies revealed deficits in the method section of all included studies. Some quality criteria were not reported, for instance, researchers' characteristics and their influence on the findings. Furthermore, barely any explicit effort to ensure trustworthiness of the qualitative data and the derived insights was reported (cf. SRQR; O'Brien et al., 2014). Overall, the quality of study reports and protocols was satisfactory. A detailed description of the quality of each study is provided in the Online Supplement 5 of the Supplementary Material.

### Discussion

The current systematic review aimed to identify and describe smartphone-delivered mental health interventions for refugee populations, thus targeting an existing knowledge gap. The review addressed the different characteristics of these interventions for refugee populations and summarized their effectiveness regarding clinical outcomes and important insights into nonclinical outcomes. The review was faced with considerable heterogeneity between interventions and study designs. It was not possible to reduce heterogeneity since the current evidence base is still sparse. Nevertheless, we were able to identify very recent literature, with all studies except one being published between 2019 and 2022. The systematic review clearly shows that various interventions have recently been developed. Currently, no smartphone-delivered intervention seems to meet the needs of the respective population comprehensively. Moreover, the specific needs of these populations might not yet be fully understood. This is reflected in ambiguous findings in qualitative interviews (e.g., regarding the advantages and disadvantages of personal contact during treatment), low utilization of apps, high dropout rates, and a lack of reliable evidence on effectiveness and efficacy.

The efficacy of smartphone-delivered interventions has been shown to be less convincing than more complex browser-based interventions, but the evidence is primarily based on Western populations (Weisel et al., 2019). The results from (pilot) RCTs included in this review reveal limited efficacy for refugee populations (Nickerson et al., 2019; Röhr et al., 2021; Lindegaard et al., 2021b; Spanhel et al., 2022), potentially for different reasons (e.g., utilization and adherence to interventions were often reported to be low). Contextual and/or cultural adaptations might be necessary. Furthermore, first evidence from non-refugee samples hint to slightly better efficacy for guided versus unguided digital interventions with at least minimal personal contact (Cuijpers et al., 2019). Although we could not find this effect clearly in the studies included, different scalable interventions for refugees from the World Health Organization (WHO) like Step-by-Step and Problem Management Plus are currently investigated in RCTs thus providing results on this question in the future (cf., Goodman et al., 2021). Recently published data on guided versions are promising (Cuijpers et al., 2022).

In addition, the systematic review did not identify any study or intervention explicitly addressing post-migration stressors. As research has indicated associations between post-migration stressors and mental health outcomes (Jannesari et al., 2020), and suggests that mental health symptoms can be effectively reduced by changing post-migration stressors (Schick et al., 2018), these should be addressed in future treatment approaches for refugee populations (cf., Goodman et al., 2021).

Trustworthiness and data security were only addressed in qualitative studies (Burchert et al., 2019) and were not mentioned in most of the study reports. Moreover, the review did not identify any data points reporting or addressing non-binary gender (with the exception of one intervention targeting only male refugees; Nickerson et al., 2019). Furthermore, while the development and adaptation of smartphone-delivered interventions are highly resource-consuming, most of the interventions were no longer available after the respective trial. The majority of data points did not report any information on availability and possible use of the respective interventions.

One strategy to address some of these deficits might be to avoid using smartphone-delivered interventions as stand-alone interventions. Several possibilities exist to implement smartphone-delivered interventions into existing mental health care systems or in contexts where no reliable mental health care system exists. For instance, smartphone-delivered interventions could be implemented as part of a stepped-care approach where they represent a low threshold intervention in a hierarchy of differently intensive interventions. Such an approach is followed by the Sanadak intervention, where results have to be awaited for future implications (Böge et al., 2020, 2022). Other smartphone-delivered interventions are used as blended-care interventions to complement group interventions (an exclusion criterion in this review; e.g., NESTT and The Happy Helping Hand) or in inpatient mental health care (e.g., *Almamar*), revealing promising initial findings (e.g., Raknes et al., 2017; Mazzulla et al., 2021). The above-mentioned scalable interventions of the WHO Step-by-Step and Problem Management Plus might offer further possibilities to combine smartphone-delivered interventions with already existing effective and scalable group interventions (e.g., Tol et al., 2020; Acarturk et al., 2022).

Most of the studies reported difficulties in recruiting the target population and high rates of dropout. Greater knowledge is required on how to attract refugees in need and how to engage them with treatment to improve their mental health and quality of life. An exception was the transdiagnostic sleep intervention, which included more participants in the pilot RCT than intended due to high demand (Spanhel et al., 2022). This might hint at the need for different approaches and a greater focus on individual symptoms rather than on general mental health.

In general, when offering smartphone-delivered interventions, it is important to consider that not all refugees in all contexts have access and conditions to use respective interventions. Beyond clinical trials, practical barriers in more naturalistic settings need to be considered when implementing these interventions. For instance, their use should be possible without a SIM card since access to SIM cards is legally limited in many countries (e.g., requirement of a valid and acknowledged identification document; GSMA, 2017). Comparably, shared mobile phones, limited access to the internet, and crowded living conditions lacking privacy need to be considered on the long term (cf., Goodman et al., 2021).

### Limitations

The findings of the current systematic review need to be considered in the light of several limitations. First, due to the broad aims of the review, the included study reports and interventions vary widely regarding several characteristics, impeding a concise summary and comparison of results. This is partly also reflected in the interrater reliability. Second, although search terms and inclusion criteria were carefully selected by several experienced researchers in the field of study, we cannot completely rule out having missed single interventions or research reports. Furthermore, since the field of study is constantly growing, new evidence is published frequently. For instance, new evidence on the intervention Step-by-Step and Sanadak has recently been published and could not be included in the systematic review (Böge et al., 2022; Cuijpers et al., 2022). Third, based on the current state of research, we were unable to look more closely at subgroups of refugees, for example, according to language, home country, or resettlement country. This might be an important topic for future research. Finally, the current systematic review does not allow for far-reaching conclusions and cannot

inform future health care decisions for refugee populations, as sufficient evidence is not yet available.

### Conclusions

This systematic review provided an overview of existing smartphone-delivered mental health interventions for refugee populations. All of the identified publications stress the importance of adequate mental health care for this highly vulnerable group. However, much more research is required on different aspects of interventions, for example, how to successfully access the target population and how to improve their treatment adherence. The benefits of smartphone-delivered interventions for this target population remain compelling, and to achieve high acceptance and utilization among refugees, it is necessary to carefully develop culturally and contextually adapted interventions with high attractiveness and trustworthiness. Moreover, to address the heterogeneity of the target population, future interventions and treatment approaches should also be as diverse as possible in order to fit the needs of more homogeneous subgroups.

**Open peer review.** To view the open peer review materials for this article, please visit http://doi.org/10.1017/gmh.2022.61.

**Supplementary material.** The supplementary material for this article can be found at https://doi.org/10.1017/gmh.2022.61.

**Data availability statement.** All data relevant to the systematic review is available within the published review and its online supplements.

**Acknowledgements.** We would like to thank Julia Lengen for her assistance with the preparation of the study report and Sarah Mannion for the language editing of the manuscript.

**Author contributions.** All authors were equally involved in the conceptualization of the systematic review. Project administration was the responsibility of R.E. and L.N. Data curation and data analyses consisting of database search, identifying gray and unpublished literature, inclusion and exclusion of data points, data extraction, and calculating interrater agreement were equally conducted by R.E. and L.N. The whole process was methodologically supervised by H.N. M.B. supervised content-related aspects and rated data points which resulted in disagreement regarding inclusion or exclusion and discussed them with R.E. and L.N. to reach a final decision. All findings were validated by M.B., C.K., and H.N. The original draft of the manuscript was written by R.E.- and L.N. It was reviewed and edited by M.B., C.K., and H.N. R.E. and L.N. shared first authorship.

**Financial support.** This research received no specific grant from any funding agency, commercial or not-for-profit sectors.

**Competing interest.** M.B. and R.E. also work at Zentrum ÜBERLEBEN gGmbH which developed and provides the intervention ALMHAR. C.K. is principal investigator and L.N. is coordinator of the research project I-REACH which develops and validates the intervention *Almamar*. C.K. is also principal investigator of the STRENGTHS project which develops and validates the Step-by-Step intervention. H.N. declares no conflict of interest.

**Ethics standards.** All authors declare to adhere to the publishing ethics of Global Mental Health.

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
