## [Reviewer Report]

Dear Editor, 

Hereby we submit the invited systematic review titled "Smartphone-delivered mental health care interventions for refugees: A systematic review". The current systematic review gives an overview of the existing stand-alone smartphone-delivered interventions for mental health problems in refugee populations. We identified nine mental health smartphone-delivered interventions for adult and three for adolescent and young refugees. The review enables the audience to identify treatments in different languages, targeting different mental problems, and offering varying amounts of support. This helps persons affected, persons working with refugee populations, and stake holders to identify the most fitting interventions for specific persons or populations. In the course of the summarized trials, about 400 refugees were provided with smartphone-delivered mental health care. The results show that the different interventions were able to improve single aspects of mental health and well-being. Still, we identified room for improvement in the efficacy and effectiveness of smartphone-delivered interventions, the involvement of post-migration stressors in the treatment, and data safety. This knowledge helps scientists and stakeholders to decide which steps should be taken next to fully exploit the potential of smartphone-delivered mental health interventions for refugees. For instance, we need more knowledge about effective treatment elements, facilitating characteristics to improve their use, and barriers which hamper the wide use in refugee populations. 

In the submission process, we had to face several difficulties: 

(1) In step 1, we were not able to select a specific issue for the invited systematic review. 

(2) In step 4, we were not able to designate Rayan El-Haj-Mohamad and Laura Nohr as shared first authors. 

(3) In general, Christine Knaevelsrud was invited to submit the systematic review. Therefore, she is logged in to the submission system. But, Laura Nohr in her role as corresponding author was submitting the manuscript. Therefore, she entered her ORCID wrongly to the name of Christine Knaevelsrud. Now, the name of Christine Knaevelsrud is linked to Laura Nohr's ORCID (= 0000-0002-3798-0909). 

(4) Also we would like to add the ORCIDs of the co-authors as well. Is there any possibility to do so?

We appreciate the opportunity of submitting the systematic review to Global Mental Health and thank you for the collaboration. 

Sincerely,

Rayan El-Haj-Mohamad & Laura Nohr

---

## [Reviewer Report]

*Comments to Author*: Row 15: consider changing the word “offer”, its meaning is not quite clear in the context.

Rows 19-21: “Moreover, adequate mental health care for refugees is associated with high costs for the host countries (e.g., due to additional costs for interpreters)” -this statement needs a reference.

Row 75: I suggest the authors spell out the abbreviation PICO directly.

Row 95: Why wasn’t the Prisma 2020 checklist used?

Rows 95-96: “The systematic review was not registered”. Although this was not done, the authors should still describe if any modifications to the search procedure were done, and if these possible modifications may have had any effect on the results of the review.

Rows 183-185: “17 full texts were considered for full-text analyses, eight study reports describing seven different interventions which met the inclusion criteria and were included in the review.” -This sentence doesn’t quite make sense, consider rephrasing.

Row 187: “Four interventions without published reports were found to be eligible and were therefore included (see Figure 1).” I think this warrants some explanation. This only serves to answer the first research question (Which smartphone delivered mental health interventions are available for refugees?) I.e. there cannot be any information regarding clinical outcomes nor non-clinical outcomes, nor information regarding dropout rates. This should be spelled out.

Rows 193-227: provide references to the cited interventions.

---

## [Reviewer Report]

*Comments to Author*: This is a straightforward systematic review on a topic that is very much

'in the picture'. The methodology is well-described and appropriate. The outcomes are to be read with some caution because research on smart-hone applications for mental health among refugees has only started. The number of studies is therefore small, but the review can provide a solid foundation for future research.

I have little to say, except some minor points:

1) when describing the interventions, it looks of use two digits after the dot when the sample is so small (n=8). this lead to suggestions of accuracy such as 41.67% and 16.67? which could be easier described as 5/12 and 2/12.

2) In the discussion I would like to read a bit more about the practical barriers to smartphone use in some settings, not just financial but also legal. for example, the Government of Bangladesh does not allow Rohingya refugees to own a Sim-card. Other issues, such as lack of networks and issues related to lack of privacy in crowded shelters could also be emphasized stronger.

3) I know that the authors need to base their findings on small number of publicly available research papers, but I would like to invite them, in the discussion and conclusion to be a bit bold. What I understand from preliminary research in refugees and from existing research in non-refugees, is that enrichment of a smart-phone app with some person to person contact greatly augments effectiveness. I think this can be better highlighted.

4) I also would appreciate if the authors could say something about how they believe smart-phone based interventions can be embedded in systems of care. They say in line 349-50 that "One strategy to address some of these deficits might be to avoid using smartphone delivered interventions as stand-alone interventions." But how could that look? Could such interventions be a first step before people go for face to face psychological interventions ? could they be combined with guided self-help intervention in groups, such as Self-hep Plus (see Tol et al 2020 and Acarturk et al 2022).

In short, the paper would become more useful if the authors in the discussion could go a bit above the direct findings...

---

## [Reviewer Report]

Dear Dr. Bass, dear Dr. Chibanda,

We appreciate the time and thoughts you and the reviewers invested to improve our manuscript before publication. In our response letter we addressed each of the comments carefully. We hope that we succeeded to your satisfaction.

Moreover, we would like to address a further topic: While revising the manuscript, we identified an additional study protocol on one of the interventions already included in the review. Thus, we are happy to provide further information on the app “BePrepared” that have not been included into the first version of our manuscript. We assume that we did not identify this study protocol because the name of the app was not unique enough to find it in the common data bases. Further, the study was registered in the German Clinical Trial Register which we did not use as source for additional reports.

Due to this additional source, we updated the results of the review and all respective documents. Therefore, the revisions of the manuscript go slightly beyond the comments of the reviewers.

Again, we thank you very much for the time and afford you spent to improve our manuscript. We are looking forward to your decision.

Sincerely,

Rayan El-Haj-Mohamad & Laura Nohr
